# Evaluating Second-Generation Deep Learning Technique for Noise Reduction in Myocardial T1-Mapping Magnetic Resonance Imaging

**DOI:** 10.3390/diseases13050157

**Published:** 2025-05-18

**Authors:** Shungo Sawamura, Shingo Kato, Naofumi Yasuda, Takumi Iwahashi, Takamasa Hirano, Taiga Kato, Daisuke Utsunomiya

**Affiliations:** 1Department of Diagnostic Radiology, Graduate School of Medicine, Yokohama City University, Yokohama 232-0006, Kanagawa, Japan; yasuda.nao.mz@yokohama-cu.ac.jp (N.Y.); d_utsuno@yokohama-cu.ac.jp (D.U.); 2Department of Radiology, Yokosuka Kyosai Hospital, Yokosuka 238-8558, Kanagawa, Japan; iwahashi.tak.th@yokohama-cu.ac.jp; 3Department of Radiology, Yokohama City University Hospital, Yokohama 232-0006, Kanagawa, Japan; hiratk@yokohama-cu.ac.jp (T.H.); t_kato92@yokohama-cu.ac.jp (T.K.)

**Keywords:** magnetic resonance imaging, image reconstruction, deep learning

## Abstract

Background: T1 mapping has become a valuable technique in cardiac magnetic resonance imaging (CMR) for evaluating myocardial tissue properties. However, its quantitative accuracy remains limited by noise-related variability. Super-resolution deep learning-based reconstruction (SR-DLR) has shown potential in enhancing image quality across various MRI applications, yet its effectiveness in myocardial T1 mapping has not been thoroughly investigated. This study aimed to evaluate the impact of SR-DLR on noise reduction and measurement consistency in myocardial T1 mapping. Methods: This single-center retrospective observational study included 36 patients who underwent CMR between July and December 2023. T1 mapping was performed using a modified Look-Locker inversion recovery (MOLLI) sequence before and after contrast administration. Images were reconstructed with and without SR-DLR using identical scan data. Phantom studies using seven homemade phantoms with different Gd-DOTA dilution ratios were also conducted. Quantitative evaluation included mean T1 values, standard deviation (SD), and coefficient of variation (CV). Intraclass correlation coefficients (ICCs) were calculated to assess inter-observer agreement. Results: SR-DLR had no significant effect on mean native or post-contrast T1 values but significantly reduced SD and CV in both patient and phantom studies. SD decreased from 44.0 to 31.8 ms (native) and 20.0 to 14.1 ms (post-contrast), and CV also improved. ICCs indicated excellent inter-observer reproducibility (native: 0.822; post-contrast: 0.955). Conclusions: SR-DLR effectively reduces measurement variability while preserving T1 accuracy, enhancing the reliability of myocardial T1 mapping in both clinical and research settings.

## 1. Introduction

Cardiac magnetic resonance imaging (CMR) has emerged as a critical tool for the non-invasive assessment of myocardial tissue properties. Among the advanced techniques, T1 mapping has shown promise in evaluating both focal and diffuse myocardial fibrosis, diagnosing heart failure and cardiomyopathies, and aiding in risk stratification [1]. Unlike traditional late gadolinium enhancement (LGE) methods, native T1 mapping and extracellular volume (ECV) mapping can detect pathological changes, including diffuse alterations that are often invisible to T1-weighted imaging, and offer a biomarker for clinical decision-making [2]. Native T1 mapping, in particular, provides a reliable assessment of myocardial damage, e.g., edema, fibrosis, amyloid deposition, and lipid or iron overload, without requiring gadolinium contrast, making it a highly reproducible tool for evaluating conditions such as cardiomyopathies [3]. Despite its utility, T1 mapping’s accuracy is susceptible to various factors, including noise, which can compromise image quality and diagnostic reliability. Not only the mean T1 value but also the histogram analysis of the distribution of T1 value may be useful in assessing the status and progression of myocardial diseases [4], and therefore, the noise reduction for T1 mapping can be important for the accurate and comprehensive evaluation of T1 mapping. Recent advancements in deep learning reconstruction (DLR) have demonstrated the potential to reduce noise and enhance image quality in CMR applications such as late gadolinium enhancement [5,6] and cardiac MR angiography [7], highlighting the opportunity to integrate deep learning-based approaches into T1 mapping for improved diagnostic utility.

The use of DLR technology has begun to be utilized as a method to enhance the image quality of cardiac MRI by improving the signal-to-noise ratio (SNR) and the contrast-to-noise ratio (CNR) [7,8]. In recent years, significant progress has been observed in traditional DLR technology, and a novel method known as super-resolution deep learning-based reconstruction (SR-DLR) has emerged. This technique, which combines denoising and high-resolution features, is gaining prominence in the field. Previous studies have demonstrated that SR-DLR significantly improves SNR and CNR in applications such as diffusion-weighted imaging (DWI) and intracranial MR angiography while maintaining quantitative accuracy [9]. Despite these advancements, the application of SR-DLR to myocardial T1 mapping has not yet been investigated.

The purpose of this research is to evaluate the impact of SR-DLR on its ability to reduce the noise of T1 mapping in CMR.

## 2. Materials and Methods

This study comprising a patient and phantom study was approved by the Ethics Committee of Yokohama City University Hospital (approval number F240700005). Written informed consent was waived due to the retrospective observational study design.

### 2.1. Patient Study Population

In this single-center observational study, we analyzed patients with known or suspected ICM or NICM who had a clinical indication of CMR at Yokohama City University Hospital (Yokohama) between July and December 2023. The study included 36 consecutive patients who underwent CMR as part of their clinical care. The exclusion criteria were severe renal impairment (eGFR < 30 mL/min/1.73 m^2^), claustrophobia, and patients with poor image quality for analysis.

### 2.2. CMR Patient Protocol

All MRI images were acquired using a 1.5-T MRI scanner (Vantage Orian, Canon Medical Systems, Otawara, Japan) and 32-channel radio frequency (RF) coils (Atlas SPEEDER Body and Atlas SPEEDER Spine, Canon Medical Systems, Otawara, Japan). Cine MRI, LGE MRI, and pre- and post-contrast T1 mapping images were obtained from all subjects. Vertical long-axis, horizontal long-axis, and short-axis cine images of the LV were acquired using a steady-state free precession sequence. A total of 0.10 mmol/kg of gadolinium contrast was injected into each patient. Approximately 10–15 min after gadolinium injection, LGE-MRI images were acquired in the same planes as the cine images using an inversion recovery-prepared gradient-echo sequence. Pre-contrast and post-contrast T1 mapping images of the LV myocardium were acquired in short-axis planes at the midventricular level using a modified Look-Locker inversion recovery (MOLLI) sequence (MOLLI 5 s [3 s]3 s; repetition time: 4.2 ms; echo time: 1.6 ms; flip angle 13°; field of view 400 × 400 mm; acquisition matrix 110 × 192). The T1 mappings underwent reconstruction both without and with SR-DLR (Precise IQ Engine (PIQE), Canon Medical Systems), utilizing identical scanning data. The PIQE technique is a super-resolution deep learning-based reconstruction method developed for MRI, aiming to generate images with high spatial resolution from data acquired at lower resolution [9]. The process involves multiple stages. Initially, an artificial intelligence (AI)-based algorithm, operating on principles similar to Canon’s Advanced intelligent Clear-IQ Engine (AiCE), selectively removes noise from both the magnitude and phase components of the image data without compromising the signal, thereby enhancing the signal-to-noise ratio (SNR). Following this denoising step, the image data are transformed into k-space, where zero-filling is applied to expand the image matrix size (e.g., up to 3-fold in-plane, resulting in a 9-fold increase in pixel count). This up-sampling step itself is not AI-driven. Subsequently, to address potential Gibbs ringing artifacts that can arise around high-contrast interfaces due to up-sampling, a second AI-based algorithm is employed to detect and reduce these artifacts while preserving the structural details. This multi-step approach, involving AI for the initial denoising, a standard mathematical process for up-sampling, and then another AI stage for artifact correction, represents a sophisticated engineering solution rather than a single end-to-end deep learning model [9]. The parameters for reconstruction without SR-DLR were as follows: matrix 384 × 384, slice thickness 10 mm, and reconstructed voxel size 1.04 × 1.04 × 10 mm and, with SR-DLR, were as follows: matrix 576 × 576, slice thickness 10 mm, and reconstructed voxel size 0.69 × 0.69 × 10 mm.

### 2.3. Phantom Configuration and Scan Protocol

Seven homemade phantoms were made by mixing Gd-DOTA and water at different dilution ratios (ranging from 1:1000 to 1:8000) and sealing them in polypropylene cylinder-shaped containers (diameter, 6 cm; height, 8 cm), so that the internal solution was at different dilution ratios (Figure 1). Coronal section imaging produced seven phantom images at a time; two types of T1 mapping images underwent reconstruction, both without and with SR-DLR. Thirty imaging sessions were performed. Static phantom scans were performed on the same 1.5T MRI system (Vantage Orian, Canon Medical Systems) equipped with an 8-channel forward array coil with these seven cylinders side by side. Data were collected using the MOLLI method, with an inversion preparation time of 110 ms, emulated heart rate at 60 bpm, and otherwise identical parameters as in the patient study (MOLLI 5 s [3 s]3 s; repetition time: 4.2 ms; echo time: 1.6 ms; flip angle 13°; field of view 400 × 400 mm; acquisition matrix 110 × 192). As in the patient study, the parameters for reconstruction without SR-DLR were as follows: matrix 384 × 384, slice thickness 10 mm, and reconstructed voxel size 1.04 × 1.04 × 10 mm and, with SR-DLR, were as follows: matrix 576 × 576, slice thickness 10 mm, and reconstructed voxel size 0.69 × 0.69 × 10 mm.

### 2.4. Quantitative Image Evaluation Methods

In the patient study, to evaluate the image noise before and after the adaptation of SR-DLR, a board-certified radiologist with 10 years of experience (S.S.) placed the regions of interest (ROIs) on T1 mapping in a mid-cavity short-axis on a medical professional imaging viewer (Ziostation2, version 2.9.7.1; Ziosoft, Tokyo, Japan) and obtained the mean, standard deviation (SD), and coefficient of variation (CV; SD divided by mean) of the T1 value before and after the adaptation of SR-DLR. The size of the ROIs was 100 mm^2^.

In the phantom experiment, each of the seven phantoms was assigned a number, and the mean, SD (i.e., image noise), and the CV of the T1 values were measured by acquiring ROIs (1957.6 mm^2^) placed in a circle inside the phantom (Figure 2).

### 2.5. Statistical Analysis

Statistical analyses were performed using the statistical software SPSS 28.0.1 (IBM, Armonk, NY, USA). All continuous data were tested for normality before analysis using the Shapiro–Wilk test and were expressed as mean ± SD or median (interquartile range (IQR)), as appropriate. The mean values of the pre-contrast (native T1 value) and post-contrast mean T1 value and SD before and after the adaptation of SR-DLR were compared using the *t*-test or the Mann–Whitney *U* test, as appropriate. Statistical significance was set at *p* < 0.05. For the quantitative evaluation, intraclass correlation coefficients (ICCs) of the mean native T1 value and mean post-contrast T1 value after the adaptation of SR-DLR by the two radiologists (S.S. and S.K.) were obtained. ICCs were used to evaluate the agreement between observers and were interpreted as follows: <0.2 = poor, 0.21–0.40 = fair, 0.41–0.60 = moderate, 0.61–0.80 = good, and 0.81–1.00 = excellent.

## 3. Results

### 3.1. Patient Study

The study included 36 consecutive patients referred for the CMR at Yokohama City University Hospital between July 2023 and December 2023. The cohort consisted of 19 males and 17 females, with a mean age of 61 ± 17 years (range 19–81 years). The most common clinical indications for CMR were investigation of the cause of heart failure and suspected ischemic cardiomyopathy. Detailed characteristics are summarized in Table 1.

Representative native and post-contrast T1 maps illustrating the image quality without and with SR-DLR are shown in Figure 3. In the patient study, the ICCs for the mean native T1 values after the adaptation of SR-DLR were 0.822 (95% CI: 0.605–0.925), while the post-contrast mean T1 values had an ICC of 0.955 (95% CI: 0.891–0.982). Both were interpreted as excellent, demonstrating high agreement between observers. For the mean native T1 values, the median (IQR) was 1000.2 ms (974.3–1013.6) with SR-DLR and 1004.5 ms (977.2–1015.8) without SR-DLR (*p* = 1.000), showing no statistically significant difference. Post-contrast mean T1 values demonstrated a median (IQR) of 556.4 ms (536.4–592.7) without SR-DLR and 563.8 ms (537.1–591.7) with SR-DLR (*p* = 0.831), also showing no statistically significant difference (Table 2 and Figure 4). For native T1 values, the mean SD without SR-DLR was 44.0 ms (95% CI: 38.6–49.4) and with SR-DLR was 31.8 ms (95% CI: 26.9–36.7) (*p* = 0.001). For post-contrast T1 values, the mean SD without SR-DLR was 20.0 ms (95% CI: 17.2–22.4) and with SR-DLR was 14.4 ms (95% CI: 12.7–16.2) (*p* = 0.001) (Table 2 and Figure 5). The mean CV also decreased following SR-DLR application. For native T1 values, the CV decreased from 4.4% (95% CI: 3.9–4.9) without SR-DLR to 3.3% (95% CI: 2.8–3.8) with SR-DLR. For post-contrast T1 values, the CV decreased from 3.5% (95% CI: 3.1–4.0) without SR-DLR to 2.6% (95% CI: 2.3–2.9) with SR-DLR (Table 2).

### 3.2. Phantom Study

In the phantom study, the mean T1 values, SD, and CV for each phantom with and without SR-DLR are summarized in Table 3. The results showed that the mean T1 values remained nearly identical across all phantoms, regardless of SR-DLR application, indicating that SR-DLR preserved T1 accuracy (Figure 6). In contrast, SD values were significantly reduced under SR-DLR conditions for all phantoms, suggesting effective noise suppression (Figure 7). Similarly, CV values were consistently lower with SR-DLR compared to without SR-DLR across all phantoms, further supporting the improvement in measurement consistency.

## 4. Discussion

This study evaluated the effect of SR-DLR on noise reduction and image quality in myocardial T1 mapping. Although SR-DLR did not significantly alter the mean native or post-contrast T1 values, it markedly reduced the SD, indicating enhanced image consistency and effective noise suppression. These findings were further supported by the phantom study, in which SR-DLR consistently decreased both SD and CV across all phantoms, without affecting the mean T1 values. Moreover, the high ICCs observed in the patient data demonstrated excellent inter-observer reproducibility with SR-DLR. Taken together, these results suggest that SR-DLR improves the reliability of myocardial T1 mapping by effectively suppressing noise while preserving measurement accuracy.

Reducing the variability in T1 measurements offers several clinically important advantages. First, it enhances the diagnostic consistency by improving the measurement reproducibility across patients and imaging sessions. This is particularly critical for monitoring disease progression and evaluating treatment response in conditions such as myocardial fibrosis—a hallmark of heart failure and other cardiomyopathies [10]. T1 mapping has been shown to predict clinical outcomes in patients with myocardial fibrosis, where higher levels of fibrosis are associated with worse prognosis [11]. By reducing the measurement variability, SR-DLR may improve the precision of fibrosis quantification, thereby contributing to more accurate risk stratification and prognostication. Furthermore, from a clinical implementation perspective, the establishment of institutional reference values typically requires sex- and age-adjusted ranges derived from at least 50 healthy subjects to detect subtle biological changes (e.g., diffuse myocardial fibrosis). However, this approach is often impractical. Improving reproducibility through SR-DLR could reduce the required sample size, thus facilitating clinical application and validation [12].

Second, the enhanced reproducibility provided by SR-DLR increases the reliability of T1 mapping as a tool for clinical decision-making. T1 mapping and ECV measurements are increasingly being used to guide therapeutic strategies, such as assessing the efficacy of interventions in cardiac amyloidosis and non-ischemic dilated cardiomyopathy (NIDCM) [13,14]. The benefits of enhanced T1 mapping precision through SR-DLR are particularly pertinent in challenging conditions like cardiac amyloidosis (CA). Native T1 values and ECV are consistently elevated in CA compared to healthy controls, reflecting interstitial amyloid infiltration [15,16]. T1 mapping can detect early cardiac involvement, sometimes even before LGE imaging shows abnormalities or clinical symptoms manifest [16,17,18]. This early detection capability is critical, and the reduced measurement variability afforded by SR-DLR can enhance the reliability of identifying these subtle initial changes. Furthermore, native T1 mapping offers a diagnostic performance comparable to LGE but without the need for gadolinium-based contrast agents [15]. This is a significant advantage for CA patients who often present with renal impairment, a contraindication for gadolinium [18]. By improving the consistency of native T1 measurements, SR-DLR can bolster the confidence in non-contrast diagnostic pathways. Prognostically, elevated native T1 and ECV values are associated with an increased amyloid burden, adverse cardiovascular events, and mortality in both AL and ATTR amyloidosis [15,19,20]. The quantitative accuracy preserved by SR-DLR, coupled with its noise reduction, is crucial for precise risk stratification. Moreover, T1 mapping is valuable for monitoring disease progression and response to therapies, such as tafamidis in ATTR amyloidosis [21,22,23]. The enhanced reproducibility provided by SR-DLR, as demonstrated in our study, can lead to more sensitive and reliable tracking of therapeutic effects, potentially allowing for the earlier assessment of treatment efficacy or disease stabilization. Thus, the integration of SR-DLR into T1 mapping protocols holds substantial promise for improving the diagnostic accuracy, prognostic assessment, and therapeutic monitoring in cardiac amyloidosis, ultimately leading to more refined patient management [24]. Otherwise, T1 mapping is known to be very useful in Anderson–Fabry disease (AFD), where it has high diagnostic potential because lipid deposition reduces native T1 [25,26]. In addition, it has been reported to be useful in monitoring drug therapies such as enzyme replacement therapy for AFD [27,28]. The ability of SR-DLR to produce consistent measurements across imaging sessions may also help refine diagnostic algorithms and optimize treatment plans for a range of cardiovascular diseases. Also, the reduced image noise in T1 mapping by SR-DLR might provide an accurate T1 value distribution, leading to an appropriate assessment of pseudo-normalization in AFD [4].

Beyond ventricular applications, the principles of improved T1 mapping precision with SR-DLR may extend to the characterization of atrial cardiomyopathy, which is increasingly recognized as a critical substrate for atrial fibrillation (AF) [29]. AF is the most common sustained arrhythmia, and while pulmonary vein isolation (PVI) is a cornerstone of ablative therapy, the outcomes, especially in persistent AF, can be suboptimal [29]. This has shifted focus towards understanding and targeting the underlying atrial substrate, particularly atrial fibrosis. LGE-MRI has been used to identify atrial fibrosis to guide ablation. However, LGE primarily visualizes a focal, dense scar and may be less sensitive to diffuse interstitial fibrosis, which also contributes significantly to the arrhythmogenic substrate [29]. T1 mapping and ECV quantification offer the potential to assess diffuse atrial fibrosis more comprehensively. The distinction between focal and diffuse fibrosis is crucial, as LGE-guided ablation targeting only visible scar might not address the full extent of the arrhythmogenic substrate, potentially explaining variable success rates in procedures like those investigated in the DECAAF II trial, where MRI-guided fibrosis ablation did not show superiority over conventional PVI [30]. While our study did not specifically investigate atrial T1 mapping, the demonstrated benefits of SR-DLR in reducing measurement variability and noise in ventricular T1 mapping are highly relevant. A more accurate and reproducible quantification of atrial T1 values and ECV, facilitated by SR-DLR, could lead to (1) improved characterization and quantification of diffuse atrial fibrosis, (2) enhanced patient selection for substrate modification strategies beyond PVI, and (3) more precise monitoring of atrial substrate changes post-ablation or in response to therapies aimed at preventing atrial remodeling. Although further studies specifically focused on SR-DLR in atrial T1 mapping are required, the potential to refine our understanding and management of the atrial substrate in AF patients is considerable.

Furthermore, reduced variability in T1 measurements benefits research by improving the precision of statistical analyses. High variability in data can obscure meaningful trends and reduce the power to detect significant effects. By minimizing this variability, SR-DLR enables a clearer interpretation of results and enhances the reproducibility of findings in clinical trials and observational studies. For example, T1 mapping has been used to track changes in fibrosis in response to treatments such as chemotherapy in cardiac amyloidosis and catheter ablation in NIDCM [14,23]. The ability to reliably measure small changes in T1 values with SR-DLR could facilitate the discovery of new therapeutic targets and improve the evaluation of treatment efficacy.

The observed reduction in noise and measurement variability with SR-DLR has several implications. While T1 mapping, particularly native T1 and ECV, excels in detecting diffuse myocardial pathologies, LGE remains the cornerstone for identifying focal myocardial abnormalities such as infarction or focal fibrosis [17]. The techniques are often complementary. Although not directly assessed in this study, the enhanced SNR afforded by SR-DLR could potentially improve the conspicuity of smaller or less apparent focal lesions in other CMR sequences like LGE, a prospect that warrants future investigation. Our institution is currently exploring the impact of SR-DLR on ECV accuracy and LGE image quality in ongoing research.

Furthermore, the application of SR-DLR may have implications for computational cost and clinical workflow. While the reconstruction process itself requires computational resources, SR-DLR offers the potential to accelerate image acquisition by allowing for a reduction in the acquired matrix size while aiming to maintain or even improve the diagnostic image quality. This could shorten scan times, benefiting patients and improving scanner throughput. However, the balance between reconstruction time, potential scan time reduction, and image quality needs to be carefully evaluated. Optimization of imaging protocols incorporating SR-DLR and comparative studies on acquisition and image processing times are also part of the ongoing research at our center.

Several limitations of this study should also be acknowledged. This study was conducted at a single center with a relatively small sample size, and further validation in larger, multi-center cohorts is necessary to confirm the generalizability of the findings. Additionally, while SR-DLR demonstrated strong performance in reducing noise, its impact on the detection of focal fibrosis and other subtle myocardial changes warrants further investigation.

## 5. Conclusions

The integration of SR-DLR into myocardial T1 mapping workflows offers significant potential to enhance the consistency, reliability, and utility of T1 measurements. By addressing key limitations associated with noise and variability, SR-DLR represents an important step forward in advancing the diagnostic and prognostic capabilities of cardiac magnetic resonance imaging.

## Figures and Tables

**Figure 1 diseases-13-00157-f001:**
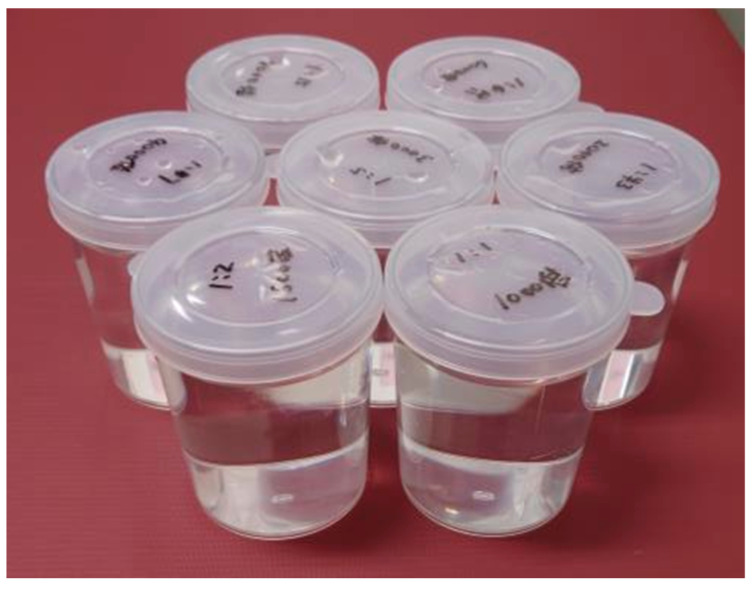
A photograph of the phantoms. Seven phantoms were made by mixing Gd-DOTA and water and sealing them in polypropylene cylinder-shaped containers (diameter, 6 cm; height, 8 cm).

**Figure 2 diseases-13-00157-f002:**
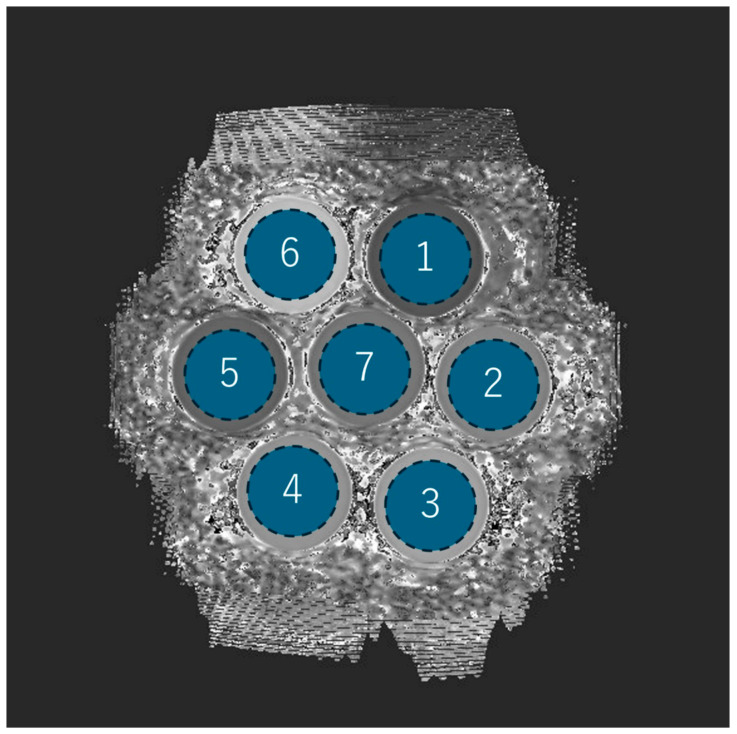
Phantom ROI setting and number assignments. Each of the seven phantoms is drawn as a circular ROI and assigned a number from 1 to 7.

**Figure 3 diseases-13-00157-f003:**
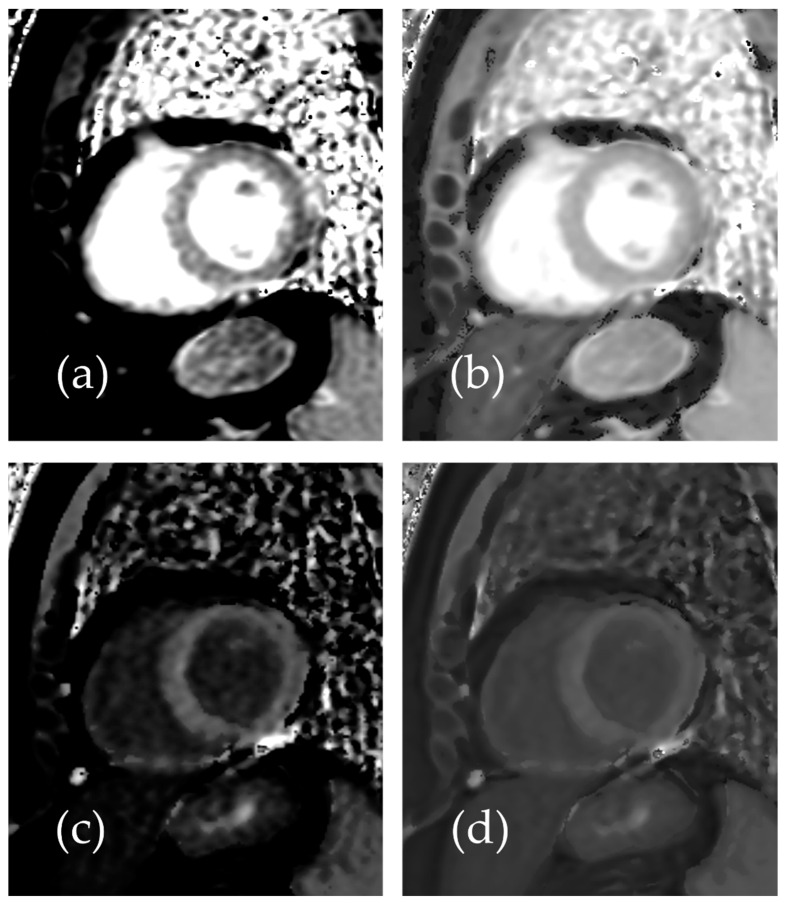
Representative images of a 54-year-old female patient with normal T1 mapping: (**a**) native T1 mapping image without SR-DLR, (**b**) native T1 mapping image with SR-DLR, (**c**) post-contrast T1 mapping image without SR-DLR, and (**d**) post-contrast T1 mapping image with SR-DLR. Images with SR-DLR (**b**,**d**) exhibit a lower noise level compared to those without SR-DLR (**a**,**c**).

**Figure 4 diseases-13-00157-f004:**
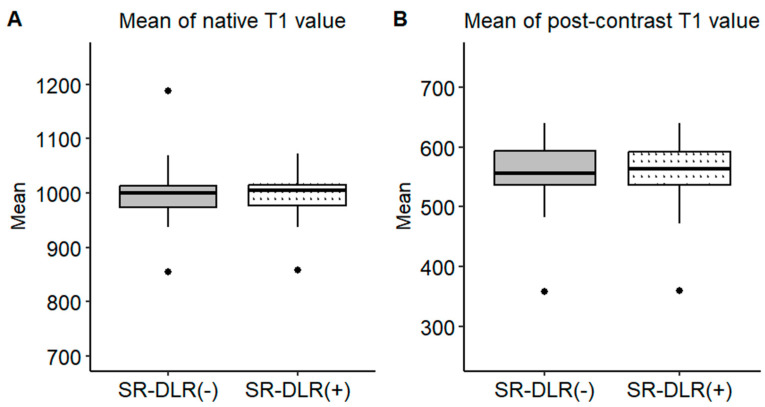
Mean T1 values in the patient study. Box plots represent the distribution of native T1 values and post-contrast T1 values before and after SR-DLR in the patient study. The median and interquartile range (IQR) are displayed for each condition. (**A**) Native T1 values before and after SR-DLR show no statistically significant difference. (**B**) Post-contrast T1 values also demonstrate no statistically significant difference.

**Figure 5 diseases-13-00157-f005:**
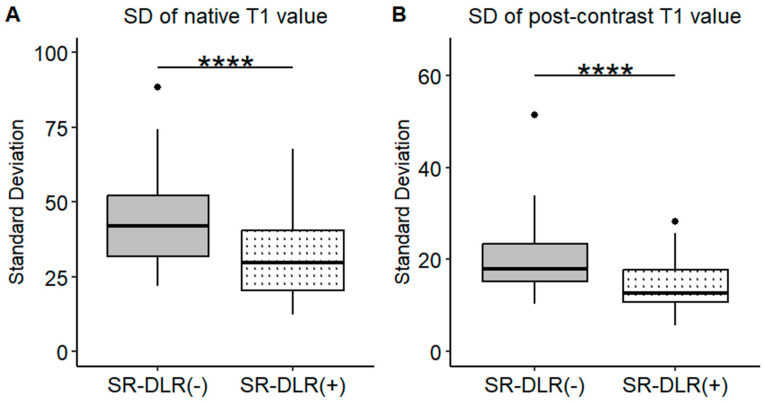
SDs of the T1 value in the patient study. Box plots represent the SDs of native T1 values (**A**) and post-contrast T1 values (**B**) before and after SR-DLR in the patient study. (**A**) Native T1 values: The SD significantly decreases after SR-DLR. (**B**) Post-contrast T1 values: The SD significantly decreases after SR-DLR. Note: ****, statistically significant difference.

**Figure 6 diseases-13-00157-f006:**
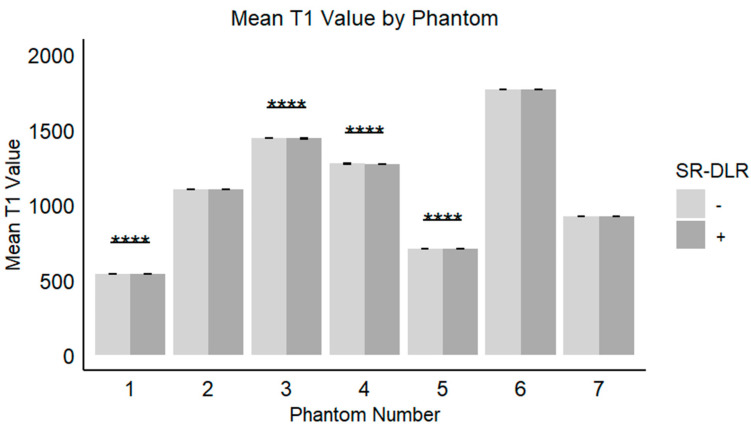
Mean T1 values in the phantom study. Box plots represent the mean T1 values without and with SR-DLR. The mean T1 values for Phantom 1, 3, 4, and 5 exhibited statistically significant differences between without and with SR-DLR conditions. Note: ****, statistically significant difference.

**Figure 7 diseases-13-00157-f007:**
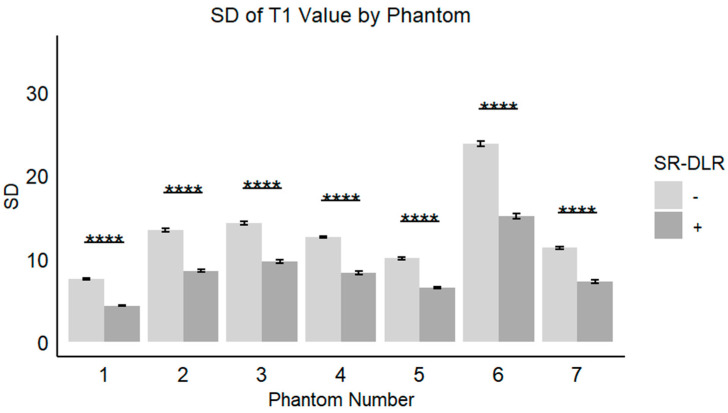
SDs of the T1 value in a phantom study. Box plots represent the SDs of T1 values without and with SR-DLR. The SDs of the T1 values for all phantoms exhibited statistically significant differences between without and with SR-DLR conditions. Note: ****, statistically significant difference.

**Table 1 diseases-13-00157-t001:** Patient’s characteristics.

Characteristics	
Patients, *n*	36
Female	17(19)
Age, mean ± SD (range), *y*	61 ± 17 (19–81)
Indication for examination, *n*	
Myocardial infarction	10
Heart failure	11
Hypertrophic cardiomyopathy	7
Dilated cardiomyopathy	2
Cardiac sarcoidosis	4
Cardiac Fabry’s disease	1
Pulmonary hypertension	1

Abbreviations: SD, standard deviation.

**Table 2 diseases-13-00157-t002:** Mean T1 value and SD at native T1 and post-contrast T1 with and without SR-DLR.

	SR-DLR (−)	SR-DLR (+)	*p*-Value
native T1			
Mean (ms)	1000.2 (974.3–1013.6)	1004.5 (977.2–1015.8)	1.000
SD (ms)	44.0 (38.6–49.4)	31.8 (26.9–36.7)	0.001
CV (%)	4.4 (3.9–4.9)	3.3 (2.8–3.8)	
post-contrast T1			
Mean (ms)	556.4 (536.4–592.7)	563.8 (537.1–591.7)	0.831
SD (ms)	20.0 (17.2–22.4)	14.4 (12.7–16.2)	0.001
CV (%)	3.5 (3.1–4.0)	2.6 (2.3–2.9)	

Note: The mean of the T1 values are presented as the median (interquartile range). SD and CV of the T1 values are presented as the mean (95% CI). Abbreviations: CV, coefficient of variation; SD, standard deviation; SR-DLR, super-resolution deep learning-based reconstruction.

**Table 3 diseases-13-00157-t003:** Mean T1 value and SD at phantoms with and without SR-DLR.

Phantom Number	SR-DLR (−)	SR-DLR (+)	*p*-Value
1			
Mean (ms)	541.9 (541.4–542.3)	541.3 (540.8–541.7)	<0.001
SD (ms)	7.6 (7.4, 7.7)	4.4 (4.3, 4.5)	<0.001
CV (%)	1.40 (1.38, 1.41)	0.81 (0.79, 0.82)	
2			
Mean (ms)	1102.5 (1101.8–1103.3)	1102.1 (1101.5–1102.7)	0.097
SD (ms)	13.4 (13.2, 13.7)	8.6 (8.4–8.8)	<0.001
CV (%)	1.22 (1.20, 1.24)	0.78 (0.76, 0.79)	
3			
Mean (ms)	1442.4 (1441.8–1443.2)	1441.6 (1441.0–1442.4)	0.003
SD (ms)	14.3 (14.1, 14.5)	9.7 (9.5, 9.9)	<0.001
CV (%)	0.99 (0.98, 1.00)	0.67 (0.66, 0.68)	
4			
Mean (ms)	1274.2 (1273.4–1274.8)	1273.5 (1272.7–11274.2)	0.019
SD (ms)	12.6 (12.4, 12.8)	8.3 (8.1, 8.5)	<0.001
CV (%)	0.99 (0.98, 1.00)	0.65 (0.64, 0.67)	
5			
Mean (ms)	707.1 (706.6–707.5)	706.6 (706.0–706.9)	0.007
SD (ms)	10.1 (9.9, 10.2)	6.5 (6.4, 6.7)	<0.001
CV (%)	1.42 (1.40, 1.44)	0.92 (0.91, 0.94)	
6			
Mean (ms)	1767.1 (1766.0–1768.4)	1767.1 (1766.0–1768.8)	0.896
SD (ms)	23.8 (23.5, 24.1)	15.1 (14.8, 15.5)	<0.001
CV (%)	1.35 (1.33, 1.37)	0.86 (0.83, 0.88)	
7			
Mean (ms)	923.4 (922.6–923.8)	923.1 (922.3–923.6)	0.221
SD (ms)	11.3 (11.1, 11.5)	7.3 (7.0, 7.5)	<0.001
CV (%)	1.22 (1.20, 1.25)	0.79 (0.77, 0.81)	

Note: The mean of the T1 values are presented as the median (interquartile range). SD and CV of the T1 values are presented as the mean (95% CI). Abbreviations: SD, standard deviation; SR-DLR, super-resolution deep learning-based reconstruction.

## Data Availability

The original contributions presented in this study are included in the article. Further inquiries can be directed at the corresponding authors.

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
