# Peer review of "Evaluating Second-Generation Deep Learning Technique for Noise Reduction in Myocardial T1-Mapping Magnetic Resonance Imaging"

_diseases, 2025, doi:10.3390/diseases13050157_

Round 1

Reviewer 1 Report

Comments and Suggestions for Authors

In their study, the authors analized the value of super-resolution deep learning-based reconstruction (SR-DLR) in increasing the accuracy of T1 mapping. The manuscript is well written, however, the authors recognized also the main limitations of the study.  I would like to ask the authors (1) to present more data about the essence of SR-DLR technique, and (2) to detail more the clinical importance of SR-DLR based T1 mapping in cardiac amyloidosis and atrial fibrillation ablation.

Reviewer 2 Report

Comments and Suggestions for Authors

The manuscript is clearly written and well-structured, with sound methodology and appropriate statistical analysis. Your findings are valuable for both clinical and research applications in cardiovascular imaging.

I would, however, like to suggest a few suggestions to enhance the clarity and completeness of the paper:

- Some p-values (e.g., p = 1.0) would benefit from clarification—does this imply exact equivalence or is it rounded? Also, including confidence intervals for SD and CV would provide additional insight into variability.

- While the boxplots are informative, including representative T1 maps (before and after SR-DLR) would add strong visual support for your findings.

- Please consider specifying the Gd-DOTA concentration ranges used in the phantoms to help readers understand how closely these reflect clinical conditions.

- Your discussion on the implications of noise reduction is excellent. It may be helpful to also address whether SR-DLR may impact detection of small or focal myocardial abnormalities, and any considerations about computational cost or clinical workflow.

Overall, this is a strong and valuable contribution, and I commend you for your work. I look forward to seeing the revised version.
